# Improved Bluetooth Low Energy Sensor Detection for Indoor Localization Services

**DOI:** 10.3390/s20082336

**Published:** 2020-04-20

**Authors:** Maja Pušnik, Mitja Galun, Boštjan Šumak

**Affiliations:** 1Faculty of Electrical Engineering and Computer Science, University of Maribor, 2000 Maribor, Slovenia; 2Foreach Solutions, 2000 Maribor, Slovenia; mitja@fsresitve.si

**Keywords:** Internet of Things, IoT, Bluetooth Low Energy, BLE, iBeacon, indoor localization

## Abstract

Advancements in protocols, computing paradigms, and electronics have enabled the development of wireless sensor networks (WSNs) with high potential for various location-based applications in different fields. One of the most important topics in WSNs is the localization in environments with sensor nodes being scattered randomly over a region. Localization techniques are often challenged by localization latency, efficient energy consumption, accuracy, environmental factors, and others. The objective of this study was to improve the technique for detecting the nearest Bluetooth Low Energy sensor, which would enable the development of more efficient mobile applications for location advertising at fairs, exhibitions, and museums. The technique proposed in this study was based on the iBeacon protocol, and it was tested in a controlled room with three environmental settings regarding the density of obstacles, as well as in a real-world setting at the Expo Museum at Postojna in Slovenia. The results of several independent measures, conducted in the controlled room and in the real-world environment, showed that the proposed algorithm outperformed the standard algorithm, especially in the environments with a medium or high densities of obstacles. The results of this study can be used for the more effective planning of placing beacons in space and for optimizing the algorithms for detecting transmitters in mobile location-based applications that provide users with contextual information based on their current location.

## 1. Introduction

The Internet of Things (IoT) field has attracted a great deal of interest in industry and academia, as it represents great potential for the development of innovative and smart solutions. The high rate of growth in the use of the IoT in our daily lives and the continuous increase in the adoption of the IoT in various smart environments (e.g., smart homes, smart universities, smart cities, and smart industry) demand the development of solutions for effective and efficient communication between IoT devices [1]. IoT networking involves balancing a set of competing requirements, such as endpoint cost, power consumption, bandwidth, latency, connection density, operating cost, quality of service, and range [2]. In recent years, the advancements in IoT protocols, computing paradigms, and electronics have enabled the development of wireless sensor networks (WSNs) with high potential for various applications in environmental, military, civil, biomedical, industrial, and other fields. A WSN is composed of a large number of sensor nodes with sensing, data processing, and communicating abilities [3]. 

Localization has been recognized as one of the most important topics in WSNs, where sensor nodes can be scattered randomly over a region and can get connected into a network [4]. Many WSN applications are location-based, leading to increasing demand for the development of new and more efficient localization algorithms. Indoor localization remains a significant and challenging task in WSNs, and it has been addressed by many researchers in the existing literature, with existing studies proposing and evaluating different solutions. A very recent survey [5] showed that, in this field, there are still open issues related to (1) localization latency, (2) efficient energy consumption, (3) accuracy and security, (4) three-dimensional localization ability, (4) environmental factors impacting the quality of the signals (e.g., obstacles, reflection, diffraction, and scattering), and (5) the cost optimization of the localization techniques. From the market for location services based on IoT, one can choose from the following main technologies: (1) Bluetooth Low Energy (BLE), (2) UWB (ultra-wide band)-based micro-location, (3) wireless positioning systems, (4) magnetic field mapping, and (5) Radio Frequency Identification (RFID) [6]. When compared to other technologies, BLE provides a fully wireless hardware solution that is autonomous in terms of power source [7]. Moreover, BLE has initiated a technology shift in the area of indoor positioning systems, and it is replacing systems based on RFID and Wi-Fi technology [8].

BLE localization systems use beacons placed at specific points in a room, providing a piece of valuable context information related to a user’s current location and surroundings that can be used for smart services such as indoor localization, issuing coupons automatically to customers who find themselves in close proximity to a store [9]. Because BLE-based location systems can be implemented and scaled easily and they do not pose a significant impact on current infrastructures, BLE technology is a popular choice for indoor locations services [7]. BLE technology is becoming more and more popular, because transmitters consume low power, are easy to use, and can be connected to smart devices such as mobiles and tablets [10]. Since recent smartphones commonly provide BLE onboard, BLE-based localization systems have become very attractive from a practical and commercial standpoint [8]. For the implementation of such solutions, BLE transmitters must be placed at specific points around the room. Mobile applications can be developed to such a degree that they are able to detect and process the data broadcast by the beacons. With the advent of BLE technology, many new opportunities and scenarios have emerged [11]. 

In existing research, several innovative solutions based on BLE-based localization systems have been proposed and evaluated, including indoor guidance and location [12], indoor positioning in a retail store [13], driver assistance in indoor parking areas [14], navigation systems for car searching in indoor parking garages [15], the real-time tracking of construction laborers [16], location-enhanced activity recognition in indoor environments [17], multi-residential activity labeling for smart homes solutions [18], location system for smart home control [19], the navigation of visually impaired people in buildings [20], and elderly people monitoring [21].

Different factors that can affect performance must be considered when designing BLE-based localization systems. In this study, we were interested in systems that use beacons for user location detection or for determining a user’s proximity to a particular transmitter. Such systems can, for example, be used in exhibition spaces, museums, castles, and other touristic buildings where the implementation of the system usually requires quick installation and the removal of transmitters. Such solutions do not focus on the user’s navigation in the area. It is more important that these solutions provide correct information based on the proximity of the user to a specific transmitter. While the user is moving in a room (or area), there is a great possibility for errors in the processing and presenting of correct information because, e.g., beacons are not transmitting the data synchronously or the mobile application can detect more than one transmitter in a specific interval. The aim of this study was to identify and analyze factors that could affect the efficiency and accuracy of detecting BLE transmitters and to improve the overall performance of a localization system. One of the aims of this study was also to improve the algorithm for user’s location detection by using the smartphone’ built-in sensors.

This study contributes to the existing literature by proposing a simple and yet straightforward algorithm based on the running average of received signal strength indicator (RSSI) values to improve the successful detection rate of the nearest beacon transmitter. A series of experiments, which were conducted in a laboratory environment as well as in a real-world environment, justified the usefulness of the proposed approach. The significance of this work lies in its experimental results, which are helpful for practitioners who wish to deploy BLE-based indoor localization services. The results of the experiments can be used as guidelines and suggestions on how to improve the localization algorithms used in context related cases, such as museums, exhibition spaces, and castles. The focus was on improving the algorithms that could be used mainly on the mobile device itself and are not heavy on computational work. Such algorithms can be used on applications that do not have a server-side communication, and all the calculations and processing have to be done on the device itself. With the results obtained from the control and real-world experiments, it was shown that a simple upgrade to track and save the previous measurements of RSSI and the use of the gyroscope to fine-tune the scanning interval can improve the overall performance of such a system. We also showed that the proposed algorithm, in conjunction with the gyroscope controlled scanning interval, did not affect battery life in any significant manner. Not only do our findings improve the quality of the systems, they also do not significantly affect the cost/time aspect of developing such systems, as we propose relatively minor and straightforward improvements. This aspect is also important for the adaptation of such systems in real-world scenarios.

The rest of the paper is organized as follows. Different techniques that can be used for implementing localization techniques based on BLE technology and a review of related work in this field are presented in the section that follows. Section 3 presents and discusses a proposed solution that addresses the challenges of existing transmitter detection techniques based on the strongest signal detection that often fail because of different factors (e.g., various physical obstacles or placement of the transmitters). Study procedures and methods are presented in Section 4, and the results of the measurements and data analysis are provided in Section 5. The results of this study, together with its main limitations and future research needs, are discussed in Section 6, and the main conclusions are provided with the last section. 

## 2. Related Work 

### 2.1. Detection and Locating Transmitters 

BLE communication consists of advertising and connecting. Advertising is a one-way discovery mechanism. Devices that need to be discovered can transmit packets of data in intervals from 20 to 2000 ms. Beacons do not broadcast the signal in intervals, not continuously. A shorter interval enables a more stable signal and, consequently, increases the accuracy of proximity approximation. However, a shorter interval impacts battery life [22]. In a BLE-based solution, a client device scans the area constantly to detect a Bluetooth signal and determine the location of the beacons. The maximum range of the signal depends on the transmission power setting and various interfering factors that can affect the broadcasting signal. Different obstacles (e.g., people and furniture) can also affect the strength of the signal, so the estimated distance is always just an approximation.

When the device detects a beacon, based on the information obtained from the beacon (e.g., signal strength and the value of major and minor values), different techniques and methods of locating can be applied to approximate the distance from the iBeacon. The distance between the transmitting beacon and the device can be estimated with the ranging process, which can result in one of four proximity states [23,24]:Immediate: This state represents a high level of confidence that the device is physically very close to the beacon.Near: This state determines a proximity of approximately 1–3 m without obstacles between the device and the beacon.Far: This state indicates that the beacon can be detected, although, because of large differences in the measured values of the transmission power, the confidence in the accuracy is too low to determine either immediate or near. This state can be caused by a number of physical obstacles between the beacon and the detecting device. This level of detection usually occurs when the device is 3–70 m away from the transmitter.Unknown (or out of range): This state represents cases when the proximity of the transmitter cannot be determined.

An RSSI value indicates the strength of the received signal that reaches the mobile device when the beacon is detected. Because beacons can broadcast their advertising packets with different transmission power (TX) values, a combination of the RSSI value and the TX power value must be used when estimating the distance to the beacon. The TX power value is the strength of the signal measured at 1 m from the device. The accuracy of the measured TX power value is crucial in calculating the distance between the beacon and the mobile device since the strength of the signal varies with the distance to the device. Knowing the RSSI at 1 m and the current RSSI (we get this information together with the received signal), we can calculate the difference using the following equation:(1)$$d=10^{\left(TX-RSSI\right)/10n}$$
where *d* is the distance in meters, *TX* is the received signal power value at 1 m, *RSSI* is the strength of the received signal, and *n* is the path loss index and relates to the environment. The value for *n* can be set based on a beacon’s location and surroundings as the following [25]: 1.4–1.9 for corridors, 2 for large open rooms, 3 for furnished rooms, 4 for densely furnished rooms, and 5 between different floors. The path loss exponent value for each beacon can also be calculated using the following equation:(2)$$n=-\left(\frac{RSSI-TX}{10\log_{10}d}\right)$$

### 2.2. Localization Techniques 

Depending on the required level of location information accuracy, Bluetooth-based localization sensing systems can vary in the number of deployable beacons, as well as in the algorithms for estimating the location of the user. For example, solutions in logistics require a large number of transmitters deployed and the implementation of more complex algorithms for providing location information with a sufficient level of accuracy. However, in the case of tourist applications that can provide visitor information relevant to their location’s surroundings, the accuracy of location information is not so important. Thus, such solutions usually need fewer transmitters and less complex location detection algorithms. 

Indoor localization algorithms based on wireless technology are divided mainly into algorithms based on the RSSI and wireless fingerprint positioning algorithms or geometric methods [26,27]. The RSSI distance-based method acquires the received signal strength from the beacon to determine the distance-loss model [26] and mostly turns the RSSI into the distance; then, through the location-distance algorithm, it estimates the user’s position [28]. There are different techniques and algorithms for determining the position of the user based on the information provided by the transmitter, and these can be classified into three main categories [29]: proximity, trilateration, and fingerprinting or scene analysis. Localization methods can also be classified into classical localization methods and artificial intelligence-based methods [30]. Classical localization methods include range-free methods (e.g., the centroid localization technique and the pattern matching method) and range-based localization methods (e.g., RSSI-based methods and time of arrival) [30]. Artificial intelligence-based localization techniques include methods such as artificial neural networks, neural fuzzy inference systems, and particle swarm optimization. Each technique has certain advantages and disadvantages, and the selection of the most appropriate one depends on the application context [13]. In existing research, studies have often combined different techniques (e.g., [19,31,32,33]) that can provide better performance.

The proximity algorithms pre-set an event-triggering threshold for a coverage area. If the RSSI values are stronger than the threshold, the target is indicated in the area [34]. The trilateration technique uses distance measurements from multiple known beacons to determine the current position of the device. If we know the distance to a transmitter in a two-dimensional world, we can conclude that the user is located at the edge of a circle, with the radius equal to the known distance from the beacon. By adding two more beacons, it is possible to get a position estimation where all the circles intersect [35]. The estimation technique (e.g., least squares or Kalman filtering) is used to estimate the target’s location from these distances and the locations of the BLE beacons [34]. Wang et al. [36], for example, used the trilateration method and applied three algorithms to estimate the location: least square estimation, three-border positioning, and centroid positioning. Fingerprinting localization methods provide a higher accuracy than radio path loss-based techniques [37]. The fingerprinting method is composed of two phases. First, all of the RSSIs of beacons at known locations are measured, processed, and stored as fingerprints in offline learning training. In the online positioning phase, the RSSI profile of a mobile device is used to produce its fingerprint, which is compared with pre-stored fingerprints to determine location by applying neural networks to determine the target’s locations [26,34,38]. 

The most basic localization technique is based on the random detection of transmitters, which is useful when an approximate estimation of location is enough. This technique is usually implemented by solutions that have to provide information related to the detected transmitter to the user or in case we want to determine if a user is in a particular room. The content shown to the user can be different, depending on the user’s distance to the transmitter. The history of positions of transmitters is not stored or used for improving the accuracy of future location estimations. Additionally, the classification of different signal strength ranges can be helpful in such detecting. If we want to show the content according to certain points of interest in a room, we can use the proximity technique, which selects the strongest signal received from a grid of transmitters. In this case, we need to scan and store information (e.g., Universally Unique Identifier (UUID), major, minor, and RSSI) of all detected transmitters at a specified time interval. After the scan is complete, we select the transmitter with the maximum power of the RSSI, assuming that the content that is attached to this transmitter is most relevant to the user. Such a method is useful when we have a large room and the points of interest in there are sufficiently distant to prevent significant disturbances in the detection of the strongest signal. In this case, a transmitter would be installed at each point of interest. If there are a lot of points of interest in the room, it is advisable to look at the financial aspect of such placement of transmitters [39]. 

### 2.3. Existing Research

The existing literature has provided a vast plethora of studies that have dealt with the performance evaluation or performance improvement of RSSI-based localization methods. BLE localization performance studies have mostly reported results from measurements in laboratory settings (e.g., [37,40,41]), although several studies have reported performance evaluation in real-world environments, such as museums [27], office environments (e.g., [14,42,43,44]), a construction site [14], a retail store [13], and underground parking [28].

Different RSSI methods, proposed by existing research, may improve the accuracy of the distance estimation to a certain extent; however, the transmission power fluctuations result in an inaccurate distance-loss model and inaccurate positioning [28]. Neburka et al. [11] evaluated the performance of the RSSI-based indoor positioning system in ideal and in real environment settings, where an increased fluctuation of RSSI values was observed in the real environment settings. To overcome the inaccurate summation of the estimated positions of the RSSI, in [12,14] a three-axis accelerometer and gyroscope were used to estimate the real-time location of the user. Rozum and Sebesta [45] proposed a new method to suppress RSSI fluctuation that uses spatial diversity (the Single-Input Multiple-Output (SIMO) approach) in combination with frequency diversity. Several studies have adopted and/or extended Kalman filter to process the Bluetooth signals and to estimate the RSSI distance model. The estimation accuracy of the RSSI distance model can be improved by implementing a backpropagation neural network (BPMN) [28], but such methods require a large amount of data to train the neural network model. Weighted least square and four-border positioning algorithms can also be used for estimating the location of the target object with a positioning accuracy that meets the requirements of the indoor positioning [27].

The performance of fingerprinting localization methods in indoor scenarios depends highly on the number of considered BLE nodes and the applied evaluation method (e.g., the number of considered sectors) [46]. Much of the existing research has been dedicated to reducing the number of referent points and the sampling time needed for fingerprint construction. To overcome these burdens, Liu et al. [42] proposed a novel approach to track a user in an indoor environment by integrating a similarity-based sequence and dead reckoning. Zhou et al. [47] proposed a semi-supervised manifold alignment approach that integrates the execution characteristic function to reduce both the number of reference points and sampling time. A fingerprinting technique based on a parallel multilayer neural network structure, which denoises RSSI directly and remembers the relationship between RSSI and position in its deep structure, can also achieve effective results in indoor positioning [37]. Wang et al. [38] proposed a novel indoor localization scheme based on subarea determination and surface fitting that enabled, on average, 10% and 22% localization accuracy improvements, respectively, when compared with the classical nearest neighbor-based fingerprinting method. Zhu et al. [40] proposed an algorithm that combines the multi-direction data collection method with standard Kalman filter and fingerprint matching algorithm to achieve the signal fluctuation reduction, noise removal, and 2D fingerprint mapping. Röbesaat et al. [17] proposed a positioning method based on fusing trilateration and dead reckoning that employs Kalman filtering as a position fusion algorithm. To improve the positioning accuracy further, they used the environmental context information while generating position fixes. To minimize the time needed for fingerprints’ collection for fine localization solutions, Danis et al. [48] used the affine Wasserstein histogram interpolation technique to approximate a radio map using a lower number of fingerprints. Recently, several studies have promoted fingerprint crowdsourcing methods to relieve the burden of site surveying [49]. Wang et al. [43] proposed a new indoor subarea localization scheme via fingerprint crowdsourcing, clustering, and matching that first constructs subarea fingerprints from crowdsourced RSS measurements and relates them to indoor layouts. They also proposed a new online localization algorithm to deal with the device diversity issue. Ye and Wang [44] proposed a scheme that contains four offline modules and one improved online positioning algorithm to deal with the challenges when constructing a radio map from crowdsourced samples. 

Combining the smartphone and the fingerprint method to determine the location is more precise and reliable, but the effort to set up the required database, especially for large buildings, is huge [6]. Jiao et al. [50] proposed a new wireless signal compensation model that considers population density, distance, and frequency. They used a convolutional neural network (CNN)-based human detection approach for estimating the number of individuals in an indoor crowded scenario and the trilateral positioning principle to realize the pedestrians’ locations. Kanaris et al. [31] combined BLE and the 802.11 infrastructure to improve the accuracy of indoor localization platforms. 

There are still plenty of challenges related to BLE technology, especially in terms of RSSI fluctuations that lead to poor precision. To mitigate these effects, Cantón Paterna et al. [51] used frequency diversity, Kalman filtering, and weighted trilateration to improve accuracy and reduce power consumption and costs in BLE-based systems. Liu et al. [19] applied a posture recognition model based on geomagnetic sensing combined with pedestrian dead reckoning (PDR) technology to solve the problems of time variation better and to improve location accuracy. Zuo et al. [32] combined the range-based method and the fingerprinting-based method, and they proposed a graph optimization-based way to estimate beacon positions and the reference fingerprinting map. Huang et al. [52] proposed an adaptable and robust algorithm based on separate channels, separate signal-attenuation models, the distance decision strategy, and weighted trilateration using BLE beacons with off-the-shelf smartphones. Wu et al. [53] proposed a way to achieve a high positioning accuracy and obtain a trajectory close to the actual path in a common application scenario with a smartphone without the use of a complicated algorithm. 

## 3. Improved Indoor Localization Based on the Strongest Beacon Signal Detection

Many existing related studies have worked on the topic of improving the overall performance of the BLE localization algorithm, but many of them have been for complex and advanced use. Such usage requires server-side calculation and is, from the cost/maintenance perspective, not suitable for context-related solutions. There are fewer experiments related to the simple and context related usage, yet this area is relatively big in terms of the practical real-world usage of such systems. Additionally, many cases of research on this similar topic were conducted in a controlled environment only. Our focus was to extensively test the proposed improvements in a controlled environment, as well as in a real-world scenario, where we deployed a solution in one of the biggest tourist attractions in Europe.

Given that we conducted performance measurements for systems used to display information that was bound to a specific area, we used the algorithm of selecting the strongest signal as a control algorithm. In the existing literature, this algorithm has been the most commonly used in such systems. The algorithm that selects the strongest signal is useful, especially when developing solutions for exhibition areas, museums, etc., because such solutions need to provide users some contextual information related to different points of interest in the area. The advantages of this technique, when compared to the others, are its quick implementation and low complexity in the placement of transmitters. However, in some cases or situations, the RSSI signal can swing heavily, causing the application to mistakenly detect the most remote transmitter as the nearest one. Such a misconception can be caused by a large number of physical obstacles in the room, or a higher transmitter density, a higher number of users in the room, among other reasons. Such errors occur more often in settings with lower scan intervals. Smaller scan intervals demand the more frequent scanning of the signals, which, consequently, increases the power consumption on the mobile device and affects the life-span of the battery. Errors in determining the correct transmitter can cause errors in displaying relevant information, which, in turn, leads to the bad user experience of visitors. A most common solution to these problems is increasing the transmitter power. However, the life-span and anatomy of the transmitter’s battery are affected by this solution. 

This study was focused on improving the technique of selecting the strongest signal from detected transmitters as follows. First, we wanted to stabilize the RSSI signal by considering previous measurements. We wanted to upgrade the existing algorithm with the ability to monitor and store the history of measurements. The input to the algorithm was, therefore, a list of all measurements taken, a list of currently detected transmitters and a number that told the algorithm how many steps back we wanted to check the measurements for each transmitter and took these values into account when calculating the new average. This improved the performance of detecting the actual closest transmitter with the sudden fluctuation of the RSSI signal or the rapid movement of the mobile device. Considering past calculations increased the stability of the algorithm, and, consequently, the mobile application was able to display the correct content to the user. 

Next, we wanted to optimize the battery consumption by implementing a smart solution that used mobile device built-in sensors (e.g., accelerometer) to automatically turn on the scanning of transmitters when the user is moving and turn off the scanning when the user is not moving. One of the biggest energy consumers of all algorithms is undoubtedly the routine of switching on and off the Bluetooth sensor, which scans the environment for signals. The scan interval and sleep interval are configurable. Our solution proposes to use the accelerometer for setting the scan interval dynamically. In case the visitor moves, the scan interval is reduced. This would increase the possibility of detecting transmitters and, thus, gain a greater number of measurements. As a result, more measurements would allow for the more accurate calculation and determination of the actual closest transmitter. 

The controlled algorithm is very basic and has no post-processing of the gathered results of detected beacons (see Algorithm 1). The algorithm requires two input parameters: scan interval and scan duration. Both parameters have to be set in milliseconds. Depending on the set scan interval value, the algorithm periodically collects data from beacons for the specified scan duration. The result of every scan cycle is an array of detected beacons. It can occur that, during the scan cycle, the algorithm detects the same beacon multiple times, depending on the transmitting settings of every individual beacon. In this case, the old value is overrode with the newer one. If this array is not empty, the algorithm loops through the results and returns the beacon with the highest measured RSSI signal strength, assuming this is the nearest beacon. In the case of an empty array, the algorithm returns null, assuming there are no beacons nearby.
**Algorithm 1** The basic algorithm**Input:** scan interval and scan duration**Output:** beacon with the highest measure RSSI signal strength1:**set** scan interval2:**set** scan duration3:**function***beacon_scanner*4: **loop**5:  **if** scan interval not over6:   **call**
*listen_for_beacons*7:**function***listen_for_beacons*8: **init** results array9: **while** scan duration not over10:  **if** beacon detected11:   **if** beacon already detected12:    override beacon data13:   **else**14:    push beacon data into the results array15:  **call**
*beacon_scanner*16:  **if** results array not empty17:   **return** beacon with the highest measured RSSI signal strength18:  **else**19:   **return** null;

Our proposed algorithm was based on the procedure that scans and returns the beacon with the strongest measured RSSI signal. However, the RSSI signal can fluctuate because of various factors (movement, dynamic obstacles, the position of a mobile phone, etc.) that can cause a sudden increase or decrease of the measured value. This can lead to false-positive or false-negative readings that cause it to display the wrong content to the user. The controlled algorithm was improved as follows (see the pseudocode for the improved algorithm in Algorithm 2). We improved the algorithm by storing detected RSSI signal strength values for each individual beacon. We did this by initializing an empty array every time we detected a new beacon and added the newly measured RSSI signal strength value to the specific array. Besides the scan interval ***s_i_*** and scan duration ***s_d_***, we had to specify the number of measurements we wanted to store for each individual beacon ***X***. For every new measurement the algorithm detects and if the maximum number of measurements that can be stored is exceeded, the first measurement in the array gets removed and the last measurement gets stored into the array, similar to the First In First Out (FIFO) principle in a fixed-length array. When a new measurement is added to the specific array, the running average for that beacon is calculated. The result of the algorithm is the strongest signal strength based on the calculated running averages for all detected beacons in the detectable area.

Another improvement in the proposed algorithm is that we took the movement factor from the user into consideration when setting the scan interval ***s_i_***. The scanning interval is set dynamically in our proposed algorithm. The maximum, minimum, and step values can be optionally for the scan interval; otherwise, default values are used. For this purpose, we developed a service listener that runs in the background and checks for movement. For movement detection, it listens for changes on the accelerometer sensor that built into the mobile device. It listens on all three axes, *x*, *y*, and *z,* for a significant change of acceleration. If a specific change in acceleration occurs (∆a), a a scan-interval update event is triggered. The threshold acceleration magnitude is set manually. The equation for detecting movement is the following:(3)$$acceleration\;magnitude=\;\sqrt{x^{2}+y^{2}+z^{2}}$$
**Algorithm 2** The proposed algorithm**Input:** scan interval, scan duration, number of measurements, acceleration magnitude threshold **Output:** beacon with the strongest signal strength based on the calculated running averages for all detected beacons in the detectable area 1:**Object***DetectedBeacon*2: **id,**3: **major,**4: **minor,**5: **rssiMeasurements[],**6: **averageSignalStrength**7:**set** scan duration8:**set** acceleration magnitude threshold9:**set** number of stored signal detections10:**init** scan interval11:**init** listOfBeacons[*DetectedBeacon*]12:**background service**13: **loop**14:  calculate acceleration15:  **if** calculate acceleration > acceleration magnitude threshold16:   update scan interval17:**function***beacon_scanner*18: **loop**19:  **if** scan interval not over20:   **call**
*listen_for_beacons*21:**function***listen_for_beacons*22: **init** results array23: **while** scan duration not over24:  **if** beacon detected25:   **if** beacon already detected26:    update rssiMeasurements array of beacon27:    calculate_running_average(beacon)28:   
**else**
29:    add beacon to listOfBeacons array30:    calculate_running_average (beacon)31:  **call**
*beacon_scanner*32:  **if** listOfBeacons not empty33:   **return** beacon with the highest calculated average RSSI signal strength value34:  
**else**
35:   **return** null;36:**function** calculate_running_average(*DetectedBeacon*)37: calculate running average for beacon38: save new calculation to specified beacon

## 4. Materials and Methods

### 4.1. Study Procedures and Methods

Before we began with the writing and planning of the experiment specifications, we researched all the possible factors and parameters that affected the beacon detections and had an influence on the RSSI signal strength readings. For more accurate and faster data collection, we developed a custom mobile application that was used to collect, analyze, and transfer the measured data to the computer for further analysis. We also used the findings and research data that were used from the Foreach Solutions company in their previous work with the development and deployment of such systems.

The experiment process for this study was composed of two phases: (1) the measurement of variables in a controlled room and (2) the measurement of the performance of the detection algorithms in the real-world environment setting. The controlled scenarios’ experiment was conducted in an empty room to first measure how the beacons impacted each other and how this interference impacted the detection process on a mobile phone (see Figure 1). All measurements in the controlled environment were conducted with a mobile phone positioned at ground level. The measurements made for detecting algorithm improvements were conducted with the phone held in a hand to simulate users moving across rooms. Additionally, all of the obstacles used in the controlled environment were custom made and built from various materials to simulate the impact of different materials and their signal reflective characteristics on impacting the detections’ process. We prepared various obstacles using wood, paper, steel, glass, and rock materials to test as many possible obstacle shapes and materials that can occur in real-world situations. The real-world experiment was conducted in the Expo museum of Postojna Caves. This environment has many obstacles that are made of various materials. We have to point out that this environment had some additional factors that affected the detection process, like visitors moving randomly around the exhibition area of the museum. 

The research model for this study was defined as follows (see Figure 2). The independent variables were the density of physical obstacles (the value varied from low to high), the density of transmitters (measured in cm), and the signal frequency (measured in ms). The observed dependent variables were signal quality (measured in dBm), the performance of the algorithm (the level of correctly detected transmitters), and the anatomy of the mobile device’s battery.

One of the aims of this study was to determine the optimal ratio between the broadcast interval and the scanning interval of the transmitter signals. Like transmitters, on a mobile device, we can specify for how long the algorithm has to scan the environment and for how long the device should stay inactive. These settings can significantly influence the battery consumption of both the transmitter and the mobile device. The impact of transmission and scanning frequency on signal quality was tested in the controlled room with one transmitter and a mobile application that was used for performing measurements. The transmitter and the mobile device were placed one meter apart. To analyze the impact of the broadcasting and scanning intervals, four independent measurements were conducted using different broadcasting and scanning configuration settings, as follows. For broadcasting, the following four intervals were: (1) 100 ms (as recommended by the Apple iBeacon Standard), (2) 250 ms, (3) 350 ms (as recommended by the manufacturer of the smart beacon), and (4) 450 ms. For detection, the following scanning frequencies were specified: (1) high frequency (HF) with scanning interval value of 500 ms and idle time interval 100 ms, (2) medium frequency (MF) with scanning interval value 1000 ms and idle time of 500 ms, and (3) low frequency (LF) with scanning interval value of 2000 ms and idle interval time of 1500 ms. Each test ran for 60 s, during which the average value of the RSSI signal, the RSSI signal fluctuation (the difference between the min and max measured values), and the standard deviation of the RSSI measurements were recorded.

The second aim of this study was to test the influence of the density of transmitters on the overall signal quality. In exhibition areas, museums, etc., there is often a case where different points of interest are very close to each other. By conducting different measurements, we wanted to find the minimum distance at which we could still successfully distinguish the transmitters from each other. To analyze the impact of the density of transmitters on the overall signal quality, we used two transmitters that were placed in the controlled environment with the following distances: 25, 50, 100, 200, and 400 cm. The level of correct detections was estimated according to the measured RSSI value. RSSI values were recorded, and average values were calculated for each measurement. The transmitter with the higher average RSSI signal value was marked as the closest one. Measurements were conducted at four locations, of which two (locations 1 and 2) were closer to transmitter 1, and the other two (locations 3 and 4) were closer to transmitter 2. The distance between the mobile device and the closest transmitter was 2 m. The measuring application was used to mark the location, enter the test distance, and run the scans. The application then performed 20 measurements and completed the scanning process. A medium frequency for the scanning interval was used on the mobile device. In total, 400 measurements, or 80 measurements per distance configuration, were conducted.

To analyze the impact of the density of physical obstacles on signal quality, three sets of measurements were conducted in the controlled room set-up, with 7 transmitters placed within a distance of at least 2 m according to previous testing results. The mobile device was placed in the middle of the space at the height that the user would hold it in his hand. The first set of measurements was conducted in an environment without physical barriers, the second set in an environment with a medium density of physical barriers, and the third set in an environment that simulated an environment with a high a density of physical barriers. Each measurement was conducted for 5 min. During the measurements, the number detections of each sensor were recorded together with the measured strength of the RSSI signal. In total, 4047 measurements were recorded. 

The performance of the standard and improved algorithm for detecting sensors was evaluated in two environmental settings: in the controlled room and in a real-world environment setting at the Expo museum in Postojna. In the case of testing in the controlled room, the measurements were conducted for the following environment types concerning the density of obstacles: a low or zero density, medium density, and high density of obstacles. In the case of testing the Expo, independent measurements were conducted for transmitters at ten locations within the museum. The transmitter detection algorithm was evaluated by navigating through the space along a predetermined path and by performing measurements of success of detection. To be able to check success of the detecting algorithm, at each pre-defined location, the tester first manually selected the transmitter closest to his current location. With selection of the transmitter, the mobile application started the algorithm for detecting the closest transmitter. After the algorithm finished, two pairs of values (the transmitter selected by the user and the transmitter selected by the algorithm) were recorded together with the measurement date and time, as well as the battery status. 

### 4.2. Testing Environment Configurations

For the controlled testing room environment, an empty room was used with the ability to add and remove things/objects, thus enabling the creation of different types of interference levels. Different materials of physical objects can affect the signal differently and are related to the absorption or reflection characteristics of materials that can cause the signal to diffuse, amplify, or block. Physical barriers within the space can greatly affect the strength of the RSSI signal that can affect the overall system’s performance. For testing purposes, three environments with different types of obstacle severity were set-up: An environment with low or no obstacle severity—a closed room without furniture or any other physical barriers (see Figure 3 and Figure 4).An environment with medium obstacle severity—a room containing some pieces of furniture and obstacles that simulated an open environment without intermediate walls, which is usually the case in trade shows, galleries, and open exhibitions. In the room (see Figure 5 and Figure 6), the following three obstructions of different materials were placed: two large wood elements (element A with a height of 1.9 m and element B with height of 2.3 m) imitating furniture obstacles, one glass element in the corner of the room, and a moderate density of people (4 persons moving randomly through the space area).An environment with high obstacle severity—to imitate the real environment and the real situation when using transmitters in an environment with many obstructions of different materials and a large number of people. This set-up was intended to replicate a high-density exhibition, museum, castle, and similar tourist attractions. In this set-up, the following obstacles were additionally placed when compared to the previous set-up (see Figure 7 and Figure 8): two large wooden obstacles imitating furniture (element F was 2.3 m high and element E was 1.5 m high) and two glass elements (elements G and D were 2.1 m high) simulating exhibition elements and other barriers placed in the room. During the measurements, 8 people were moving randomly through the space area.

Both sensors’ detection algorithms were also tested at the Expo Museum, which is part of the Postojna Cave. The Expo Museum provides visitors a detailed view of the Karst world that forms and surrounds the Postojna Cave [54]. Visitors are also provided with a mobile app that enables an audio-guide through the exhibit by displaying content related to specific points of interest (see Figure 9). The displayed content and notification within the app is triggered by sensing the location of the visitor and the Bluetooth transmitters installed throughout the museum. 

The Expo Museum (see Figure 10) is an example of a very dynamic space with a high density of obstacles of different materials (e.g., metal obstacles, pillars, and various exhibition objects). There is a lot of technological emphasis on presenting the topics to visitors. The broadcast interval on the transmitters was set to 150 ms, and all transmitters were installed at 2 m or higher (see Figure 11).

### 4.3. Testing Equipment

The “smart beacons” manufactured by Kontakt (see Figure 12) were chosen for deployment because they provide a good balance between battery autonomy and quality of operation in terms of detection and the ability to adjust different parameters (reach, signal strength, etc.) [8]. In the experiment, 10 transmitters were configured with a unique UUID to each transmitter. The major value of all transmitters was set to 42. The minor value was used to identify each transmitter in the area, where the values ranged between 1 and 10. According to the manufacturer’s recommendation, the transmitter power was set to 4, which allowed for a −8 dB transmission signal strength and a maximum range of 30 m. CR2477 batteries were used for the power supply. The table below (see Table 1) shows an overview of the transmitter configuration.

This study was focused on the use of Android-based mobile devices, because the Android operating system offers more configurable options compared to the iOS operating system with limited possibilities. The Android operating system enables access and configuration of the Bluetooth sensor (e.g., it is possible to set, when the sensor is activated, how long it will take to scan the environment for other devices, what happens to the data acquired, etc.). The mobile device used for testing was a Samsung Galaxy S7 with Android 7.0, 3000 mAh, Li-Ion battery, Bluetooth 4.2, A2DP, LE, aptX, with 4GB RAM.

For measurement purposes and for collecting data during the experiments, a mobile application was developed and used as a diagnostics tool to detect and measure all the critical factors required to perform the analysis and comparison of detection algorithms proposed in this paper (see Figure 13). The developed diagnostic application enabled automated and more efficient data measurement, because we were able to collect more data over a shorter period of time. The diagnostic application was composed of five modules, as follows. 

An RSSI signal strength scanning module was the main module that starts by running the application. The main goal of this module is to gather as many RSSI signal power readings in as short a period of time as possible. The scanning process starts by entering the distance between the beacon and the mobile device running the application, and by clicking the button “Start scanning.” The scan procedure then scans for all iBeacon standard signals periodically and saves detected signals (every positive reading) into a temporary result database (“Cache”). From every positive reading, we saved the UUID, major, minor, and RSSI signal strength values.

The algorithm comparison module was developed and used to test the accuracy of the control and proposed algorithm for beacon detection. While moving through the room, the module enables manual selection and automatic detection of the nearest beacon based on the calculation. The procedure of this module is as follows. First, the tester selects the type of algorithm, where there are only two possibilities: the control algorithm and the proposed algorithm. After the algorithm is selected, the tester has to press the button “Start test” to start the algorithm. As the tester is moving through the room, he also has to manually select the beacon that is actually the closest to him. After the manual selection, automatic detection is triggered using the selected algorithm. After the algorithm calculates the nearest beacon, the manual and calculated values are stored in a temporary database (“Cache”). The procedure is finished by pressing the button “End test.” After the test has ended, the measured data can be exported for further data analysis. 

The closest beacon detection module enabled us to measure how the distance between beacons affected their detection. Because all beacons transmit their signal asynchronously, it is important to know the minimum distance between installed beacons, which still enables correct detection. A data export module was built to enable easy and quick export of all the gathered data over various communication channels (upload to the cloud, send via email, etc.). The export module enables data export in a Comma-Separated Values (CSV) format. During signal detection and measurements, the mobile application also stores information about the battery status and the consumption of the battery. A battery recording module was also developed to enable this. When the app is launched, this module starts as a background service and is responsible for reading the battery status every X seconds, where the X value is configurable. This service reads and saves the battery status and timestamp of the reading periodically. To increase the reliability of the battery consumption data before conducting measurements, the mobile device was reset to factory settings, the sim card was removed, and Wi-Fi connections were disabled. These actions were performed to exclude, or at least minimize, the effect of other applications or services on battery consumption during running the measuring procedures.

## 5. Results

### 5.1. The Impact of the Transmission and Scanning Frequency on Signal Quality

The measurements of the signal quality in relation to transmission and detection frequency (see Table 2) provided different results for different transmission intervals and detection frequency configurations. The worst result was measured when the transmission interval was set to 450 ms and a high detection frequency was used. Using a medium scan frequency and a 250 ms transmission interval enabled the most stable, as well as the strongest, RSSI signal. Though higher scanning frequencies allowed for more measurements, the quality of the signal was poor, with higher fluctuations in RSSI values. High fluctuations can lead to a false prediction or detection of the nearest transmitter. The results also showed that the low-frequency scans could also enable satisfactory results in the signal quality. A combination of a low scan frequency and a 450 ms transmission interval would also be possible, especially in cases where users move through space at a slower pace.

### 5.2. The Impact of the Density of Transmitters on Signal Quality

The tests that measured the success of transmitter detections considering the density of the transmitters showed the influence of the asynchronous signal (see Table 3). The transmitters in these trials were not interconnected, and therefore their signal intervals were not coordinated. The results of testing showed that the RSSI oscillation was mainly a problem in settings with transmitters that were placed very close. Distance of 25 and 50 cm were too small and did not allow for the effective distinguishing of transmitters. The success rate of correct detection was 50% or less, which means that the result of the detection of the transmitters placed apart with a small distance was practically random. At a distance of 100 cm, the success rate of detection improved slightly but was still 62.5% and insufficient. When planning the installation of transmitters, a distance of at least 200 cm must be considered. Namely, at a distance of 200 cm, we measured an 87.5% success of correct detection. At a distance of 400 cm, the success of correct detection was even higher, with a value of 98.75%.

### 5.3. The Impact of the Density of Physical Obstacles on Signal Quality

When analyzing the impact of the density of physical obstacles on the correct detection of transmitters, the measurements showed that adding more physical objects did not significantly affect the number of detected transmitters. The numbers of detected sensors recorded were very similar for all transmitters in all three environment settings (see Table 4). However, differences were observed in the perception of the signal strength concerning the material of the physical barriers, as well as the position of the barriers themselves. For transmitter 3, an anomaly was observed, meaning that the average RSSI signal strength was increased by adding more barriers to the room. The reason for this could have been the placement of a physical barrier C with a glass material, which could have reflected the signal and increased the strength of the signal. The same phenomenon was observed for transmitter 6, for which the strength of the signal was also increased with an increased level of obstacle severity and was probably caused by the glass material of the physical obstacle G. 

The measurements also showed that physical barriers could impact the strength of the RSSI signal (see Table 5). The smallest differences in the measured signals were observed for transmitter 2, which was an expected outcome, since in all environment settings, there were no barriers between transmitter 2 and the device that was used for measuring. The obstacles that were placed directly in front of the transmitters had the strongest impact on the RSSI signal strength. The measures also showed signal strength amplification when adding glass obstacles into the room, which caused signal reflections. 

### 5.4. Performance Evaluation of the Proposed Algorithm

Overall, the measurements provided evidence that the proposed algorithm performed better in all three scenarios for the density of obstacles in the controlled room (see Figure 14, Figure 15 and Figure 16 and Table 6). The measurements for transmitters 5 and 6 showed that both algorithms had difficulties in detecting the correct transmitter. The reason for this was in the small distance between the two transmitters, which caused problems in signal detection and, consequently, affected the algorithm for selecting the closest transmitter due to the RSSI signal fluctuation.

In the controlled room, our improved algorithm for detecting and finding the closest transmitter performed better; the success rate was 14.29% higher compared to the success rate of the standard algorithm (see Table 7). The results also showed that the proposed algorithm performed better, especially when detecting in a space with a medium or high density of physical obstacles. In the case of a higher density of physical barriers, the success rate of our algorithm was 20% higher compared to the results provided by the standard approach. The battery consumption measurements provided very similar results for both algorithms, and no significant difference was observed. The results showed a slight improvement in power consumption (0.04%) when the proposed algorithm was used.

The results of the standard and improved algorithm performance measurements showed that the proposed algorithm also performed better in the real-world environment setting (see Figure 17 and Table 7). The existing algorithm achieved an overall 87% success rate of correct transmitter detection. The accuracy of our proposed algorithm in a real-world setting was 95%, which was 8% higher when compared to the performance of the standard algorithm. The proposed algorithm considers calculated averages of RSSI signals from past measurements, which contributes to a more stable total value of RSSI signals and, consequently, a lower probability of detecting incorrect or more distant transmitters. The differences in battery consumption while running both algorithms (standard algorithm vs. proposed algorithm) were insignificant.

## 6. Discussion

Several factors that can influence the beacon’s signal quality must be considered when designing a solution based on BLE technology. A beacon’s signal quality in a room with multiple transmitters can be affected by (1) the density of transmitters or the distance between individual transmitters, (2) the density of physical barriers and the characteristics of the barriers, and (3) transmission and scanning frequencies. Regarding the configuration of the transmitters and scanning devices, it was shown that minimal fluctuations and, consequently, the best signal quality can be achieved when the beacon’s transmission interval is set to 250 ms and the device is set with medium scanning frequency (1000 ms scanning interval and 500 ms idle interval time). Low-frequency scans can also enable a satisfactory quality of the signal; however, such configuration is applicable only for cases where the scanning device can or must move at a slower pace throughout the area. A short distance between transmitters, which are placed in the room, can also lead to RSSI oscillation. The best signal quality providing a higher level of detection accuracy can be achieved with transmitters placed 2 m or more apart. For the best results, it was shown that the distance should be at least 4 m to provide the best success rate in the detection algorithm. The barriers can also affect the signal quality, as the barriers that are placed directly in front of the transmitters weaken the signal strength. However, the signal strength can also be amplified if obstacles made of glass are placed near the transmitters.

The algorithm proposed in this study uses a series of measurements of the RSSI signal to be able to select the sensor based on the strongest signal more accurately. Different factors, such as a high density of obstacles and deployed transmitters and a lower transmitter power, can influence RSSI signal quality. A bad signal quality can lead to false results in detecting the transmitter closest to the scanner device location. The results of several independent measures conducted in the controlled room, as well as in the real-world environment, showed that the proposed algorithm for detecting the closes transmitter outperformed the standard algorithm in different environmental conditions, especially in the environment with a medium or high density of obstacles. The success rate of the proposed algorithm was higher by 14.29–20% in the controlled room setting and 8% in the real environment setting in the Expo Museum in Postojna. One of the biggest energy consumers is the routine of switching the Bluetooth sensor on and off. To optimize battery consumption, in this study, a dynamic setting of the scanning interval was proposed that, based on the analysis of the data provided by the accelerometer, can dynamically configure the scanning interval in real-time according to the speed of the user’s movement. The battery consumption measurements provided very similar results for both algorithms, and no significant difference was observed. The results showed a slight improvement in power consumption (0.04%) when the proposed algorithm was used.

As in all research, this study has some limitations that need to be identified and discussed to make the findings more objective. First, the equipment used for designing and implementing the network of beacons was limited to the selected hardware. Therefore, the transmitters’ broadcasting capabilities, as well as the configuration options, were limited to the abilities of the selected product. Further trials are needed to test the proposed technique using other transmitter devices. Next, the scanning device in this study was an Android-based mobile device, which provides more possibilities regarding the configuration of the scanning device and the implementation of the procedures using the BLE standard protocol. Future research is needed to test other mobile devices and operating systems that enable implementing solutions based on the BLE standard. Future research needs to include other BLE standards as well. The proposed algorithm should be tested with devices that are based on BLE protocols like Eddystone and others.

There is a possibility for bias in the results because the measurements were limited to the environment settings with obstacles and people as designed by this study. It is of special note that it was very hard to recreate the same conditions regarding the non-fixed or random movement of people in the room. In order to reduce the risk of validity and reliability, several safety steps were included in the study. In the controlled room, all visitors knew their exactly pre-defined positions and paths along which they had to move during the measurements. However, in the Expo Museum, there was no possibility to control random visitors’ movements during individual measurements. Though the proposed algorithm performed better in all conditions, the variability of the results was higher for the real-world environment. Finally, the measurements were conducted with a specially developed application that enabled us to minimize the bias in the accuracy of measurements.

## 7. Conclusions

This study contributes to the body of research in the field of design and development of Bluetooth-based localization sensing systems. These systems vary in the number of deployable beacons, as well as in algorithms for estimating the location of the user. This study especially focused on algorithms that can be applied in systems that display information to the user when approaching a specific point in the area. The main objective of this study was to test and improve the most common algorithm used in existing applications that selects the strongest signal. The algorithm that selects the strongest signal is useful, especially when developing solutions for exhibition areas, museums, etc., because such solutions need to provide users some contextual information related to different points of interest in the area.

The main novelty of this study is the proposed improved algorithm, which can detect the closest transmitter based on the data from previous measurements. Based on the analysis of the data from previous measurements of the transmitter signals, the performance of detecting the actual closest transmitter was improved. By including the data from a built-in accelerometer, the scanning algorithm was optimized for battery consumption, enabling the scanning of transmitters to start only when a device is being moved. The results of experiments carried out in this study are also important, because they can be used as guidelines for better placement of transmitters and their configurations for optimal broadcast intervals according to the density of physical barriers and the expected density of visitors. The results of this study can help practitioners in the cost and timing aspects of the overall development of location-based solutions for visitors to exhibitions, exhibition grounds, or museums.

This study revealed a lot of potential for future research. Testing on the transmitters themselves should be highlighted as starting points for further research. The biggest issue that is not affected by current protocols and operating mode is asynchronous signaling. The overall detection efficiency and performance could be improved if transmitters were able to coordinate transmission intervals with each other. This could create a synchronous network of transmitters and enable developing a Bluetooth mesh [55].

## Figures and Tables

**Figure 1 sensors-20-02336-f001:**
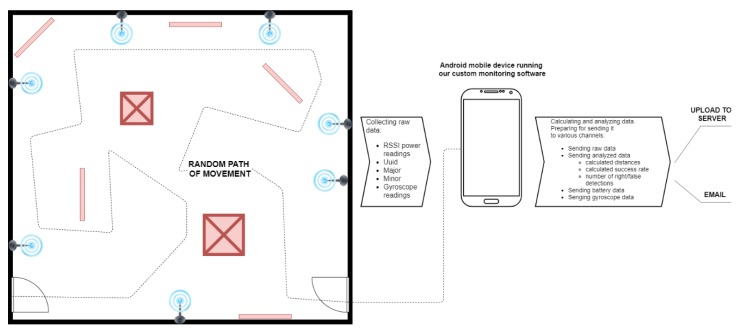
The flow of experiment measures in a controlled room.

**Figure 2 sensors-20-02336-f002:**
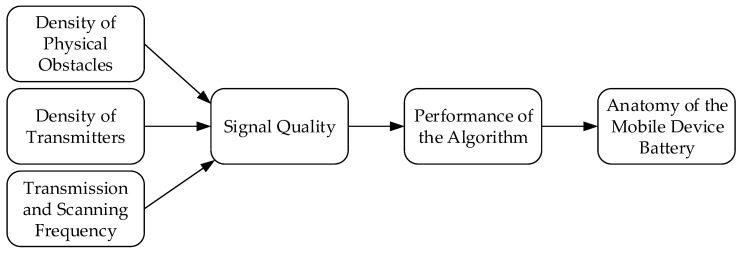
The conceptual research model.

**Figure 3 sensors-20-02336-f003:**
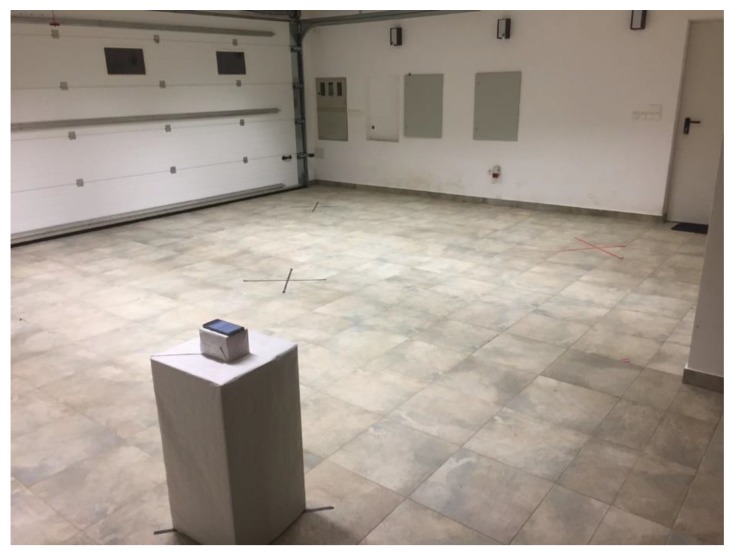
Testing environment with low or no obstacle severity.

**Figure 4 sensors-20-02336-f004:**
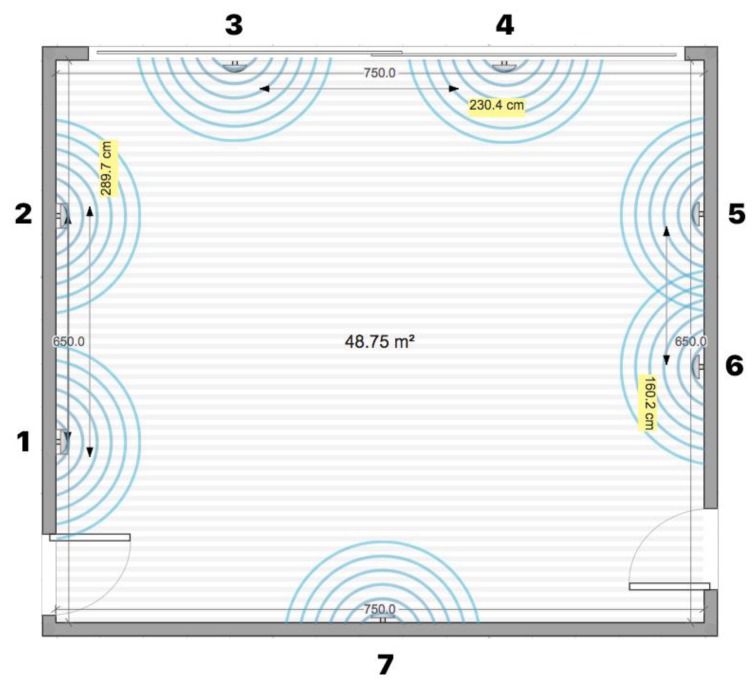
Testing environment with low or no obstacle severity.

**Figure 5 sensors-20-02336-f005:**
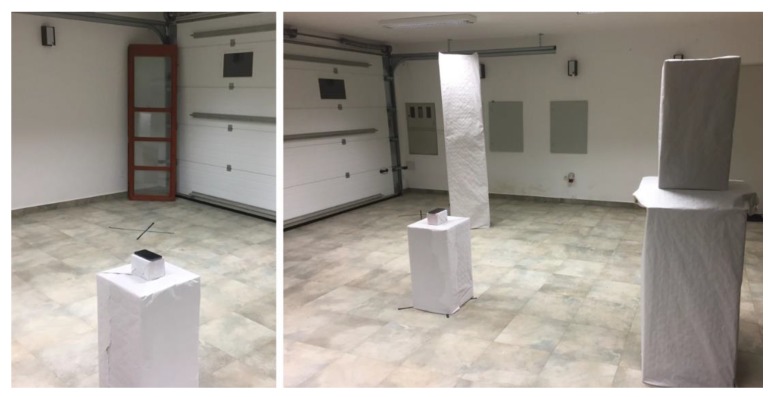
Testing environment with medium obstacle severity.

**Figure 6 sensors-20-02336-f006:**
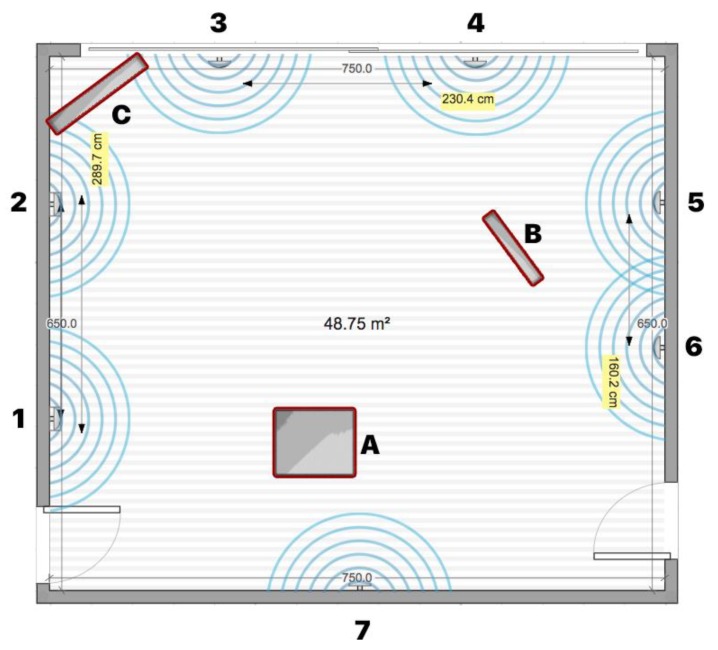
Testing environment with medium obstacle severity.

**Figure 7 sensors-20-02336-f007:**
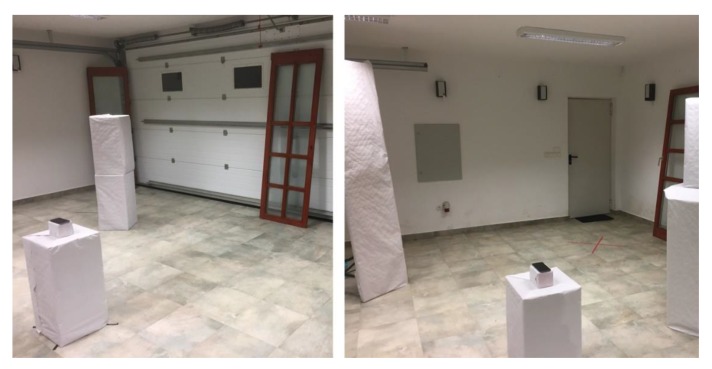
Testing environment with a high level of obstacle severity.

**Figure 8 sensors-20-02336-f008:**
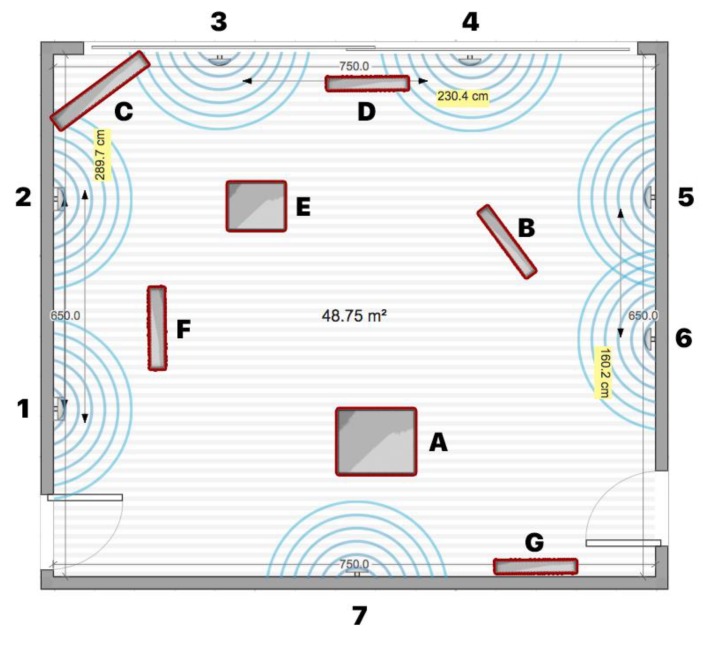
Testing environment with a higher level of obstacle severity.

**Figure 9 sensors-20-02336-f009:**
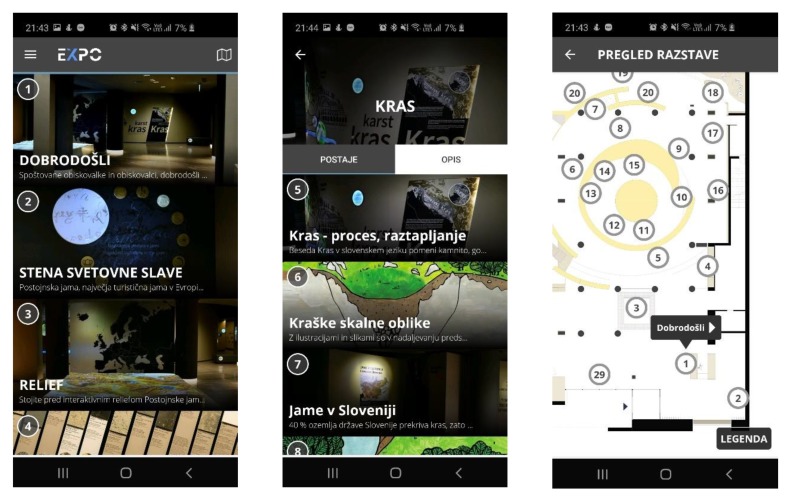
Screens of the Expo app for visitors (in the Slovenian language).

**Figure 10 sensors-20-02336-f010:**
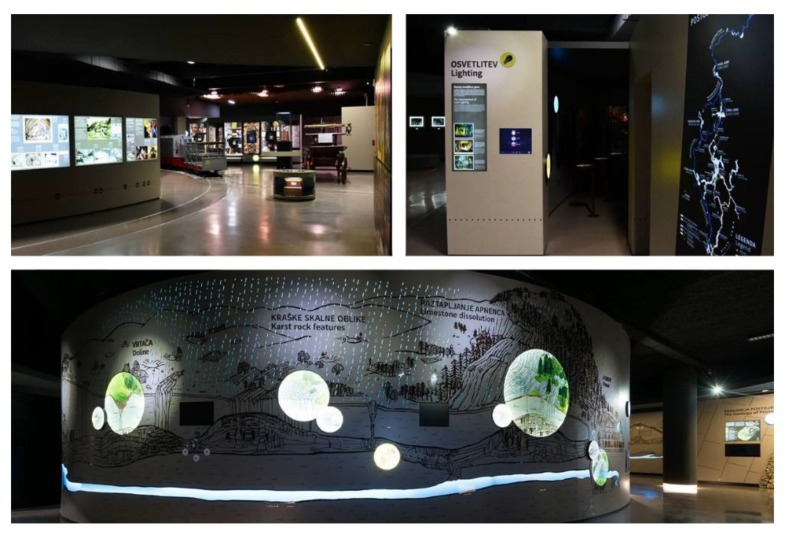
Expo Museum.

**Figure 11 sensors-20-02336-f011:**
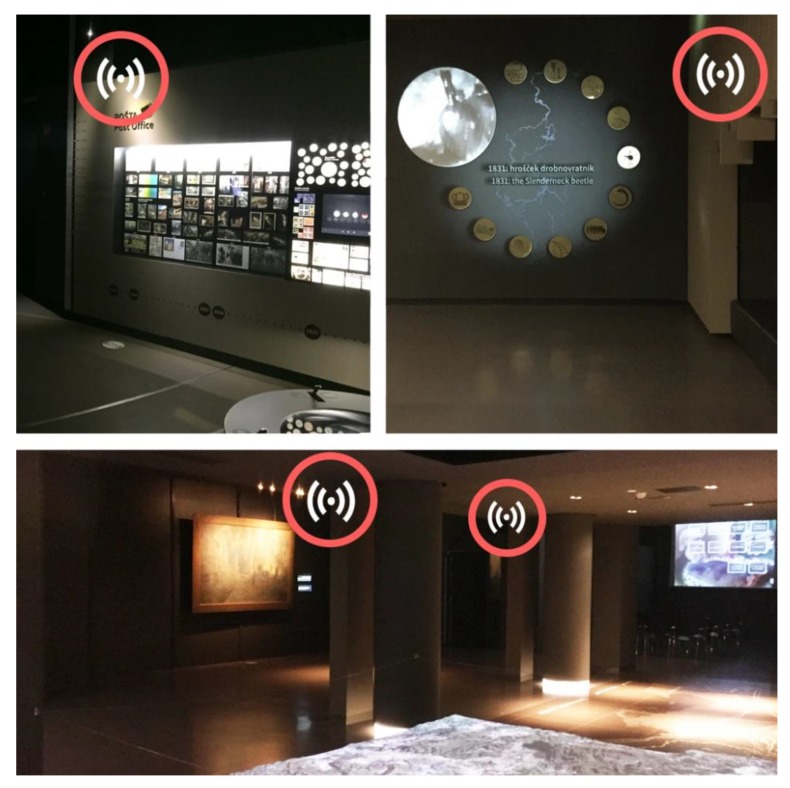
The locations of smart beacons in the Expo museum.

**Figure 12 sensors-20-02336-f012:**
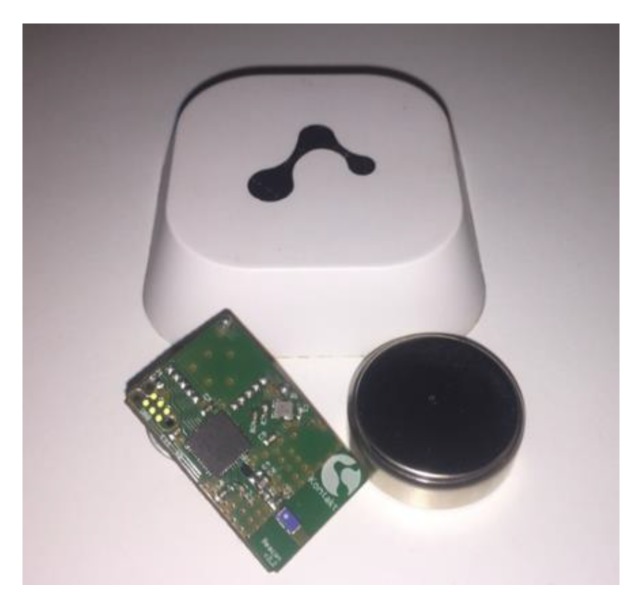
The smart beacon.

**Figure 13 sensors-20-02336-f013:**
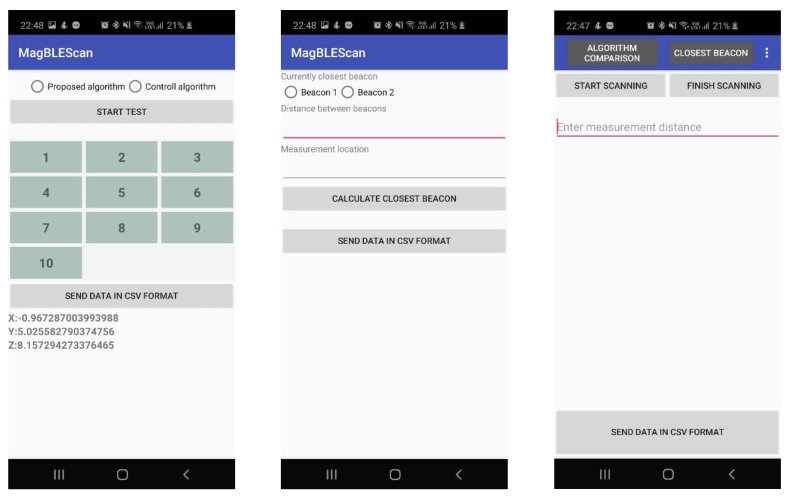
Screens of the diagnostic application used for measurements and data collection.

**Figure 14 sensors-20-02336-f014:**
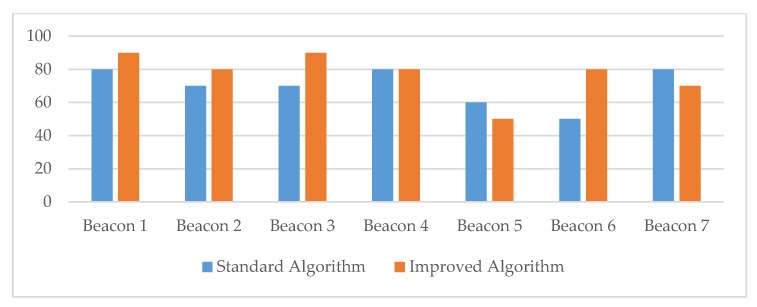
The success rate of correct transmitter detections for the controlled room environment with a low or no density of obstacles.

**Figure 15 sensors-20-02336-f015:**
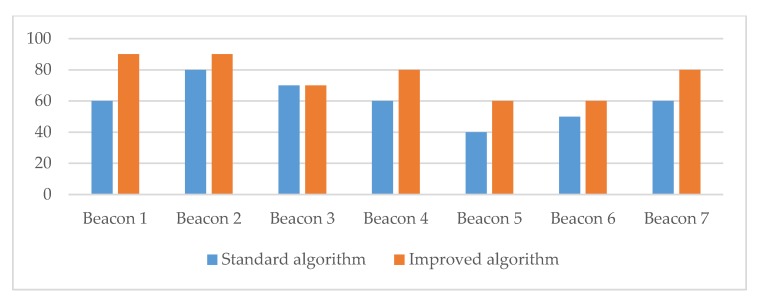
The success rate of correct transmitter detections for the controlled room environment with a medium density of obstacles.

**Figure 16 sensors-20-02336-f016:**
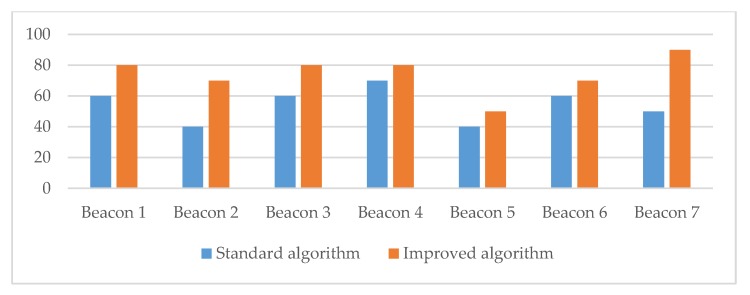
The success rate of correct transmitter detections for the controlled room environment with a high density of obstacles.

**Figure 17 sensors-20-02336-f017:**
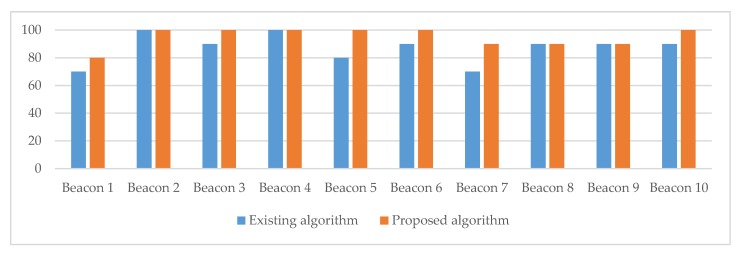
The success rate of correct transmitter detections at the Expo Museum, Postojna.

**Table 1 sensors-20-02336-t001:** The configuration of the transmitters.

UUID value	4c414d41-4fa2-4e98-8024-bc6a6e7a6b6d
Major value	42
Minor value	1–10
Battery	CR2477 3V
Transmitter power	0–7

**Table 2 sensors-20-02336-t002:** Signal quality measurements for different transmission and scanning interval configurations. RSSI: received signal strength indicator.

		Detection Frequency [Scanning Interval, Idle Time]		
Transmission Interval	Signal Quality	High [500 ms, 100 ms]	Medium Frequency [1000 ms, 500 ms]	Low Frequency [2000 ms, 1500 ms]
**100 ms**	**RSSI Average**	−85.96 dB	−79.45 dB	−82.33 dB
	**Standard Deviation**	2.53	2.75	2.27
	**RSSI Signal Fluctuation ^1^**	13	11	10
**250 ms**	**RSSI Average**	−84.78 dB	−76.23 dB	−78.55 dB
	**Standard Deviation**	2.67	1.22	2.85
	**RSSI Signal Fluctuation ^1^**	11	4	12
**350 ms**	**RSSI Average**	−81.91 dB	−81.47 dB	−80.89 dB
	**Standard Deviation**	4.65	2.8	2.13
	**RSSI Signal Fluctuation ^1^**	19	11	8
**450 ms**	**RSSI Average**	−85.59 dB	−77.73 dB	−80.55 dB
	**Standard Deviation**	5.47	1.84	1.09
	**RSSI Signal Fluctuation ^1^**	21	9	3

^1^ RSSI signal fluctuation is the difference between the maximum and minimum RSSI value.

**Table 3 sensors-20-02336-t003:** Number of correct transmitters detected in relation to the distance between two transmitters and the location of the receiver.

	Distance between Two Transmitters				
	25 cm	50 cm	100 cm	200 cm	400 cm
**Location 1**	2	17	4	20	20
**Location 2**	17	15	14	18	19
**Location 3**	0	2	17	14	20
**Location 4**	16	6	15	18	20
**Summary**	35/80	40/80	50/80	70/80	79/80
**Level of correctly detected transmitters**	43.75%	50%	62.5%	87.5%	98.75%

**Table 4 sensors-20-02336-t004:** Number of correct detections of transmitters in relation to the density of physical obstacles.

Density of Obstacles	Number of Success Detections (Number of Measures = 200) Beacons						
	1	2	3	4	5	6	7
**No or low**	192	193	190	189	195	191	187
**Medium**	192	188	191	195	195	194	195
**High**	194	196	192	195	193	193	197

**Table 5 sensors-20-02336-t005:** Average measured RSSI signal strength (measured in dB) in relation to the density of physical obstacles.

Density of Obstacles	Average Measured RSSI Signal Strength Beacons						
	1	2	3	4	5	6	7
**No or low**	−77.31	−81.83	−86.95	−82.72	−81.6	−81.81	−78.75
**Medium**	−78.90	−82.21	−83.43	−85.14	−80.57	−84.62	−79.23
**High**	−81.55	−82.76	−82.14	−85.86	−82.40	−83.89	−80.57

**Table 6 sensors-20-02336-t006:** The success rate of correct transmitter detections for different environment settings (controlled room).

	Success Rate of both Detection Algorithms in Relation to the Density Level of Obstacles Beacons						
	1	2	3	4	5	6	7
**Low or no density**							
Standard algorithm	80%	70%	70%	80%	60%	60%	80%
Improved algorithm	90%	80%	90%	80%	50%	80%	80%
**Medium density of obstacles**							
Standard algorithm	60%	80%	70%	60%	40%	50%	60%
Improved algorithm	90%	90%	70%	80%	60%	60%	80%
**High density of obstacles**							
Standard algorithm	60%	40%	60%	70%	40%	6%	50%
Improved algorithm	80%	70%	80%	80%	50%	70%	90%

**Table 7 sensors-20-02336-t007:** Comparison of overall success rates of both algorithms.

	Existing algorithm	Improved Algorithm	Level of improvement
**Controlled environment**			
No or Low Density	71.42%	78.57%	+7.15%
Medium Density	60%	75.71%	+15.71%
High Density	54.28%	74.28%	+20%
Overall success rate	61.9%	76.19%	+14.29%
**Real-world Environment**			
Success rate	87%	95%	+8%

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
