# Peer review of "Improved Bluetooth Low Energy Sensor Detection for Indoor Localization Services"

_sensors, 2020, doi:10.3390/s20082336_

Round 1

Reviewer 1 Report

Section 1 does not show what is the paper's contribution.

Or how the authors achieved that contribution.

A lot of the paper is dedicated to describing BLE that is an already well-known technology.

In the fragment:

"This study is focused in improving the technique of selecting the strongest signal from detected
transmitters as follows. First, we want to stabilize the RSSI signal by considering previous
measurements. We want to upgrade the existing algorithm with the ability to monitor and store the history of measurements."

This is already well-known and done in the wireless sensor networks fields. There should be a broader discussion of related works in order to see the localization field with more depth.

Doing it in a real world environment it was good. It was a really nice experiment.

Overall it seems a really nice case paper, but in its state it is difficult to see the contribution. There is no comparison whatsoever with any other approach in the literature. And the literature review is shallow and it does not point out the relevance of the work.

The experimental part is really interesting.

Reviewer 2 Report

The authors proposed a simple and yet straightforward algorithm based on the running average of RSSI values to improve the successful detection rate of the nearest beacon transmitter, and conducted a series of experiments to justify the usefulness of their approach. I think that the significance of this work lies in its experimental results, which are helpful for practitioners who wish to deploy BLE-based indoor localization services.

Here are some suggestions for the authors to improve the quality of the paper:

  1. Fig. 2 "preffix": Should be "prefix".
  2. Line 367 (Fig. 5) "if scan interval over": Should it be "if scan interval not over"? Please confirm. 
  3. Line 395 "FILO": To me it seems to be "FIFO" given that the first measurement in the array gets removed first, so it is out of the array first.
  4. Fig. 7 "Signal Qualty": Should be "Signal Quality".
  5. Line 467 "signal frequency": This term is ambiguous.
  6. Line 476 "(LF) with interval value": Should be "(LF) with scanning interval value".
  7. Line 628 "can configure": Should be "is configurable".
  8. Line 646 (Table 2): Please give a formal definition of "Oscillation of the RSSI signal".
  9. Section 5 could be reorganized into subsections, and use each subsection to describe one specific type of experiment. 
  10. Section 2 could be trimmed to reduce the total length of the manuscript.

Reviewer 3 Report

This paper proposed an improved method to detect the closet transmitter for BLE. The results of this paper are of practical significance for the application of BLE positioning. The reviewer has the following comments for this paper:

  1. This paper introduces too much background of IOT and WSN. The author should reduce unnecessary parts in order to increase the readability of articles.
  2. All equations should be numbered specifically, such as (1).
  3. The author does not introduce the specific algorithm of Bluetooth localization in recent years.
  4. The proposed algorithm should be written in the form of pseudo code in the table.
  5. The author should state the process of the experiment in detail.
  6. The author should introduce some references about the improved standard method.
  7. The author should illustrate the innovation of the proposed method than the other improved standard method.

Round 2

Reviewer 3 Report

1.The author adds a broader discussion of related studies to discuss the localization field in more depth in order to enrich the content of the article. Meanwhile, the experiment is done very well. 2.The annotation format at the bottom of the picture should be centered.

Author Response

We would like to thank the Reviewer for the time and efforts that were needed for a thorough review of our manuscript. We very much appreciate all the comments and suggestions, which we read carefully, and used to improve the manuscript. Below, we provide point by point responses to each general/specific comment that the reviewers have made to the original manuscript.

Comment 1: The author adds a broader discussion of related studies to discuss the localization field in more depth in order to enrich the content of the article. Meanwhile, the experiment is done very well.

Response: We would like to thank the reviewer for the comment and kind recognition.

Comment 2: The annotation format at the bottom of the picture should be centered.

Response: According to the suggestion, we centered the annotations at the bottom of all figures.